# Antibacterial Properties In Vitro of Magnesium Oxide Nanoparticles for Dental Applications

**DOI:** 10.3390/nano13030502

**Published:** 2023-01-27

**Authors:** Adriana-Patricia Rodríguez-Hernández, Alejandro L. Vega-Jiménez, América R. Vázquez-Olmos, Miriam Ortega-Maldonado, Laurie-Ann Ximenez-Fyvie

**Affiliations:** 1Laboratorio de Genética Molecular, División de Estudios de Posgrado e Investigación de la Facultad de Odontología, Universidad Nacional Autónoma de México (UNAM), Ciudad de México 04510, Mexico; 2Laboratorio de Bioingeniería de Tejidos, División de Estudios de Posgrado e Investigación de la Facultad de Odontología, Universidad Nacional Autónoma de México (UNAM), Ciudad de México 04510, Mexico; 3Instituto de Ciencias Aplicadas y Tecnología, Universidad Nacional Autónoma de México (UNAM), Ciudad de México 04510, Mexico

**Keywords:** nanoparticles, magnesium oxide, mechanochemical, antibacterial, bacteria biofilm resistance

## Abstract

(1) Dental caries, periodontitis, or peri-implantitis are commensal infections related to oral biofilm former bacteria. Likewise, magnesium oxide nanoparticles (MgO-NPs) were studied to introduce them to the antibacterial properties of a few microorganisms. Considering this, the purpose of the present investigation was to determine the antibacterial properties of MgO-NPs on representative oral strains. (2) Methods: MgO-NPs with a cubic crystal structure were obtained by magnesium hydroxide mechanical activation. After synthesis, the MgO-NPs product was annealed at 800 °C (2 h). The MgO-NPs obtained were tested against ten oral ATCC strains at ten serial concentrations (1:1 20.0–0.039 mg/mL per triplicate) using the micro-broth dilution method to determine the minimal inhibitory concentration (MIC) or minimal bactericidal concentration (MIB). Measures of OD595 were compared against each positive control with a Student’s *t*-test. Viability was corroborated by colony-forming units. (3) Results: The polycrystalline structure had an average size of 21 nm as determined by X-ray diffraction and transmission electron microscopy (high resolution). Antimicrobial sensitivity was observed in *Capnocytophaga gingivalis* (MIB/MIC 10–5 mg/mL), *Eikenella corrodens* (MIB 10 mg/mL), and *Streptococcus sanguinis* (MIB 20 mg/mL) at high concentrations of the MgO-NPs and at lower concentrations of the MgO-NPs in *Actinomyces israelii* (MIB 0.039 mg/mL), *Fusobacterium nucleatum* subsp. *nucleatum* (MIB/MIC 5–2.5 mg/mL), *Porphyromonas gingivalis* (MIB 20 mg/mL/MIC 2.5 mg/mL), *Prevotella intermedia* (MIB 0.625 mg/mL), *Staphylococcus aureus* (MIC 2.5 mg/mL), *Streptococcus mutans* (MIB 20 mg/mL/MIC 0.321 mg/mL), and *Streptococcus sobrinus* (MIB/MIC 5–2.5 mg/mL). (4) Conclusions: The MgO-NPs’ reported antibacterial properties in all oral biofilm strains were evaluated for potential use in dental applications.

## 1. Introduction

The use of magnesium oxide as a nanomaterial (MgO-NPs) has increased, especially in the medical field, such as in diagnostics, detection, and biosensors of molecular behavior [1,2]. The antibacterial effect of nanoparticles based on magnesium oxide has shown effective properties on a wide range of microorganisms, for whom the antibacterial mechanism of action is due to the size [2,3] and dosage dependence [4,5]. There are different approximations to explain the antibacterial mechanism of MgO-NPs. The interaction of nanoparticles with bacteria increases the production of reactive oxygen species inside bacteria cells and Ca^2+^ ions concentrations. Additionally, the interaction of MgO-NPs with specific molecular sites of bacteria in the planktonic state can trigger membrane disruptions, leading to bacteria death and the cytoplasm bacteria alkalinization of pH from 7 to 10 with high concentrations of Mg^2+^ ions [5,6]. Other toxicity effects of MgO-NPs indicate the potential number of reactive groups dependent on superficial oxygen and the weak points on the bacterial surfaces [3,5]. Despite these studies, and unlike other metal oxide nanoparticles, there is not much information on MgO nanoparticles in biological and antimicrobial applications [2,7].

Therefore, it is important to deepen the study of MgO-NPs and to look for new forms of synthesis for their possible oral biological applications, given the excellent chemical and biocompatible properties of MgO [7,8,9]. Likewise, dental biofilm is compounded by around 400 species, with representative genera *Actinomyces*, *Capnocytophaga*, *Eikenella*, and *Streptococcus* [10,11,12]. Some of them are strongly involved in biofilm formation conjoint with dental problems such as caries or periodontitis. For example, *Streptococcus mutans* is considered an antimicrobial target for caries. Nevertheless, other acidogenic and lactic acid producers are strongly associated with dental demineralization, including *Lactobacillus* sp., *Actinomyces* sp., and *Veillonella* sp., among other cariogenic bacteria [13,14,15]. Furthermore, there is an association between periodontal diseases and a pathogenic biofilm compound made up of a high proportion of certain genera, such as *Porphyromonas, Prevotella*, and *Fusobacterium* [16]. On the other hand, oral and body commensals like *Staphylococcus aureus* are considered some of the most resistant strains and biofilm formers in medical devices. Additionally, the literature demonstrates their role in nosocomial and oral infections such as peri-implantitis [17,18,19].

Applications of nanoparticles in the dental field are possible, and the materials can be used for oral-disease preventive antiseptics and dental biomaterials to improve their properties and the quality of treatments [20,21,22]. Thus, some reports have evaluated the obtention of nano-material available for sale by testing the toxicity effects in humans. [23,24,25]. Therefore, it is necessary to seek no toxic elements that are also biocompatible for clinically safe use. In this work, we present MgO-NPs obtained by the mechanochemical method to evaluate their antibacterial activity against representative bacteria related to biofilm formation and recognized as commensal strains involved in oral infections such as dental caries or periodontal diseases.

## 2. Materials and Methods

### 2.1. Nanoparticles Materials

Magnesium hydroxide tetrahydrate, Mg(OH)_2_·4H_2_O (98%), and acetone CO(CH_3_)_2_ (99.5%) were purchased from Sigma-Aldrich (St. Louis, MO, USA). All elements were used without further preparation. Ultrapure water (ddH_2_O, 18 MΩ/cm) obtained from a Barnstead E-pure deionization system has been used.

### 2.2. Synthesis and Characterization of MgO-NPs

The synthesis was performed and described in a previous study [26], using a chemical reaction to obtain MgO-NPs expressed like this:$$\mathrm{Mg}{\left(\mathrm{OH}\right)}_{2}\;\frac{\mathrm{Milling}}{800\;{}^{\circ}C/2\;h}\to \;{\mathrm{MgO}}_{\mathrm{NPs}}+{\;H}_{2}O$$

The X-ray diffraction patterns had been carried with radiation (CuKαλ = 1.5406 Å) in a diffractometer D5000 Siemens; the intensity of diffraction was analyzed as being between 2.5° and 70°, with 2θ steps of 0.02°, for 0.8 s each point. MgO-NPs’ average size (D) was calculated using their diffractograms by the Debye–Scherer equation, D = κλ/βcosθ, where κ the shape factor is equal to 0.9, CuKα radiation is λ, β is the full width at half the maximum intensity of the selected peaks (FWHM), and the Bragg angle is θ. FT-IR scanning was performed on the molecular materials in the form of KBr pellets and on films deposited on silicon wafers using a Nicolet spectrometer. Transmission electron micrographs were performed by transmission electron microscopy (high resolution), obtained through a JEOL 2010 FasTEM analysis microscope (Tokyo, Japan) at 200 kV by depositing a drop of the powdered MgO-NPs dispersed in ethanol on 300 mesh Cu grids coated in a carbon layer.

### 2.3. Antibacterial Susceptibility Testing

The bacterial species applied were acquired from the American Type Culture Collection (Rockville, MD, USA) from lyophilized stocks. The antibacterial susceptibility test was performed in a certified quality management system ISO: 9001:2015 from the Molecular Genetics Laboratory, at the dentistry school at UNAM.

The selection of oral commensals was based on resistance biofilm strains, and early and later biofilm colonizers, described in Table 1, all represent dental biofilm bacteria. All of the strains were cultured in an anaerobic chamber with an 80% N_2_, 10% CO_2_, and 10% H^2^ anaerobic environment, rehydrated in *Mycoplasma* broth base, and cultured in trypticase soy agar (TSA) enriched with 5 mL hemin, 0.05%, 500 mL distilled water UV/UF, 5 mL of menadione 0.005%, and 25 mL of defibrinated ram blood at 35 °C for 3–7 days. Each strain was harvested from TSA in pure cultures and resuspended in 1.8 µL microcentrifuge tubes with trypticase soy broth (TSB) enriched as well as TSA, and the optical density (OD) of each strain was read with the Eppendorf® Uvette® in a UV–visible spectrophotometer (Eppendorf) with each reader at an optical density λ = 600 nm (OD600). Every strain was adjusted to one to obtain 10^9^ cells/mL. Then decimal 1:1 dissolution was performed to obtain a concentration with 10^6^ cells/mL, and the solution was then transferred to 96-well clear polystyrene microplates (Corning®, Glendale, AZ, USA) for microbial susceptibility testing.

The bacterial susceptibility testing of oral strains was evaluated by the micro broth dilution test. The ten serial dissolutions (1:1) from 20 to 0.039 mg/mL of the MgO-NPs were tested against each evaluated bacteria (100 µL at 10^6^ cells). Then 100 µL of each dispersion was added per well for a final volume of 200 µL. For the negative control of the inhibition it was used the TSB culture media with each strain, and then amoxicillin was added (1 mg/mL). The microplates of each bacteria strain were exposed to the MgO-NPs incubated in an anaerobic environment as previously described at 35 °C in an orbital that was shaken at 160 rpm for 72 h, except for *P. gingivalis* and *P. intermedia* with 120 h culture and additionally enrichment media with hemin, 0.05%. After incubation, each microplate was analyzed for absorbance reading. The absorption was measured in a spectrophotometer ultraviolet–visible (UV–Vis) Filter Max F5 multi-mode microplate reader; a wavelength of 595 nm was taken as the standard value for bacteria. An extra plate was done with all the serial dispersions for the MgO-NPs (in the absence of bacteria) with the purpose of subtracting the absorbance’s nanoparticles. For the determination of growth inhibition, to confirm cell viability after the susceptibility test, we recovered an aliquot per dispersion evaluated and performed an anaerobic re-culture from 5–7 days in a TSA-enriched solution to determine the resistance or susceptibility of each tested bacteria by the counting of colony-forming units (CFUs).

### 2.4. Data Analysis

Each triplicated assay was averaged, and data results were compared for each positive bacteria control with the Student’s *t*-test (IC at 95%) in SPSS software. For strict statistics for the results, the analysis was adjusted with multiple comparisons as follows, described in the formula *p* of 0.05 = 1 – (1 − *k*)^3^ [27], *k* = using the individual *p*-value of 0.0168.

A colony counter and a stereomicroscope were used for each re-culture sample in TSA enriched for the viability test by CFUs (Fisherbrand™). The interpretation of bacteriostatic sensitivity by minimum inhibitory concentration (MIC) was determined by the counting of visible colonies; the minimum bactericidal concentration (MBC) was reported when there was no growth, with CFUs = 0 and the resistance evaluations reported as CFUs = + (positive and uncountable growing) or as non-visible inhibition as previously reported [28].

## 3. Results

### 3.1. Characterization of MgO-NPs

The X-ray diffraction patterns obtained from the MgO-NPs show a face-centered cubic (fcc) lattice with the space group *Fm3m* (Figure 1a). The peaks observed were indexed according to the Joint Committee on Powder Diffraction Standards (JCPDS) with the card number: 89-7746, and the corresponding crystal planes of MgO were indexed as (111), (200), (220), (311), and (222). The peaks in the XRD patterns corresponding to the single crystalline phase and the average crystallite sizes were calculated by the Scherrer equation, with the crystal plane at 200; the estimated size was 21 nm. Respecting the FT-IR spectrum (Figure 1b), the signals at 3697 cm^−1^ and 3436 cm^−1^ are characteristic of the O–H stretching band. The peaks at 1449–1591 cm^−1^ correspond to the bending vibration of the O–H; these strong bands appear due to the hygroscopic nature of MgO, whereas the strong peak around 455 cm^−1^ is due to the stretching vibrations of the bond between Mg and O in MgO-NPs. HRTEM micrographs (Figure 1c,d) indicate the shapes to be polyhedric, with sizes smaller than or close to 100 nm. Likewise, the analysis results in high resolution showed that the interplanar distances coincide with the reflections of the MgO crystal structure, as observed in the XRD results (Figure 1a).

### 3.2. Antibacterial Susceptibility Testing

The microbial susceptibility of biofilm species is described in Table 2, with the OD595 values, the Student’s *t*-test significance after the micro broth dilution assays under ten MgO-NPs dissolutions (20–0.039 mg/mL), and the CFUs’ reported recovery of viability. The MgO-NPs exhibited bactericidal (MIB) and/or bacteriostatic (MIC) effects on all biofilms’ former strains.

The lower sensitivity to the MgO-NPs at the maximum concentration evaluated (20 mg/mL) was in the biofilm-resistance strains *S. aureus* and *S. mutans.* However, both presented bactericidal (CFUs = 0) or bacteriostatic effects (CFUs = 15), respectively. *S. mutans* presented bacteriostatic sensibility (MIC) at lower concentrations (0.312 mg/mL, non-significant, NS, and CFUs = 0). Additionally, *S. aureus* presented an MIC of 2.5 mg/mL (NS, recovery of viability CFUs = 0) (Table 2 and Figure 2A) even though it had not presented a MIB at any dissolution.

*S. sanguinis* (Table 2 and Figure 2B) (early colonizer), with one of the later colonizer biofilm strains *P. gingivalis* or a periodontopathogen strain (Table 2 and Figure 3B), reported no recovery of viability with CFUs = 0 until the maximum concentration evaluated (20 mg/mL). However, *P. gingivalis* presented bacteriostatic sensibility at a lower concentration of 2.50 mg/mL (NS, CFUs = 300).

The strain with the opposite result, the most sensitivity to MgO-NPs, was the early colonizer *Actinomyces israelii,* with an MIB at 0.039 mg/mL (*p* < 0.01, CFUs = 0) (Table 2 and Figure 2B).

There was sensitivity to MgO-NPs at intermediate concentrations against early colonizers, such as *Capnocytophaga gingivalis,* with MIB at 10 mg/mL (*p* < 0.05, CFUs = 0) and MIC at 5 mg/mL (*p* < 0.05, CFUs = 6); *Eikenella corrodens,* with MIB at 10 mg/mL (NS, CFUs = 0) (Table 2 and Figure 3A); and *Streptococcus sobrinus,* with MIB at 5 mg/mL (NS, CFUs = 0) and MIC at 2.5 mg/mL (NS, CFUs = 3) (Table 2, and Figure 2B).

The highest susceptibility to the MgO-NPs was reported against the later colonizers or putative pathogenic strains, such as *Fusobacterium nucleatum* subsp. *nucleatum,* with MIB at 5 mg/mL (NS, CFUs = 0) and MIC at 2.5 mg/mL (NS, CFUs = 10), and *Prevotella intermedia* with MIB at 0.625 mg/mL (NS, CFUs = 0) (Table 2, and Figure 3B).

## 4. Discussion

The characteristics of MgO-NPs by mechanosynthesis are consistent with other studies that include the results of FT-IR [29,30,31] and HRTEM [32]. These characteristics contribute to the antibacterial properties of nanoparticles based on MgO affecting bacterial viability, especially due to nanoparticle size and concentration [2,3,33].

Concerning oral strains playing the role of biofilm formers as cariogenic or periodontal pathogens [11,16], the antimicrobial effect of MgO-NPs has been previously reported [2,3,7], as has as its cell biocompatibility at low concentrations [34]. However, the antibacterial properties of nanostructures based on MgO itself against subgingival strains have been poorly evaluated, finding little or no information [35,36,37].

Our study analyzed the effect of MgO-NPs without any other component, showing good antibacterial properties against cariogenic bacteria, such as *S. mutans*, against periodontopathogenic bacteria or later colonizers, and against *S. aureus,* one of the resistant biofilm strains of medical devices and commensal infections in peri-implantitis [17,18,19]. On the other hand, MgO NPs presented less sensitivity to early colonizers of subgingival biofilms, such as *C. gingivalis*, *E. corrodens*, and *S. sobrinus*, suggesting a selective antibacterial effect against pathogenic strains. The only exception was the *A. israelii* strain with the highest sensitivity to the MgO-NPs. Despite the benefits of *A. israelii* being an early colonizer of the dental biofilm, it has been a recognized commensal pathogen related to dental caries [13,15] and actinomycosis with odontogenic-originating infections [38]; hence, their higher sensitivity to MgO-NPs could be a positive feature to introduce in dental materials.

Another selective antibacterial effect was on the cariogenic strain *S. mutans* over the less cariogenic bacteria *S. sanguinis*. Competition between acidogenic and lactic acid bacteria continuously coexists in the dental caries process [12]. In this case, if the MgO-NPs had presented more sensitivity for *S. mutans* than *S. salivarius*, it could promote a less cariogenic microbiota in infants. Herein is the suggestion to apply MgO-NPs to dental materials before *S. mutans* infectivity until children are three years old or before the window of infectivity by the cariogenic strain is closed [39].

However, only one study reported a significant antibacterial effect on a recent cement, a glass ionomer material based on MgO-NPs, in the biofilm activity of cariogenic bacteria [36]; in this case, the glass ionomer cement has an antibacterial effect itself, and the nanoparticle size is similar to our study. Other dental material compounds with MgO-NPs promote MgO nanocellulose membranes introduced for periodontal tissue regeneration [40] with antibacterial properties against strains such as *E. coli* and *A. actinomycetemcomitans*. Some other dental types of cement modified with zein-MgO nanoparticles showed significant antimicrobial properties against the fungal species *C. albicans* and the bacteria strain *S. aureus* [35]. In the present study, the best antibacterial susceptibility was observed in MgO-NPs themselves, without combination with another material or substance, on periodontal or putative pathogens or later colonizers of the subgingival biofilm *P. gingivalis, P. intermedia,* and *F. nucleatum* subsp. *nucleatum*.

The application of nano-composed dental materials may be a solution to disturb biofilm formation [41] in commensal infections, such as caries, periodontitis, or peri-implantitis, and also as a therapeutic tool to combat microbial resistance, for example, periodontal treatments coadjutant as antiseptics, since the resistance of used antiseptics or antibiotics has no selectivity to periodontal pathogen strains [42]. The present research results proposed the possibility of introducing these MgO-NPs to antiseptics or dental materials due to the antibacterial sensitivity to cariogenic strains such as *S. mutans*, to periodontal pathogens as later colonizers of dental biofilm, and to biofilm-resistant strains such as *S. aureus*.

## 5. Conclusions

The MgO-NPs had reported antibacterial properties in all oral biofilm strains evaluated for potential use in dental applications due to the antibacterial sensitivity at low concentrations for the cariogenic strain *S. mutans* over *S. sanguinis*, for the periodontal and putative pathogens over early biofilm colonizers, and for *S. aureus,* at the evaluated concentrations.

## Figures and Tables

**Figure 1 nanomaterials-13-00502-f001:**
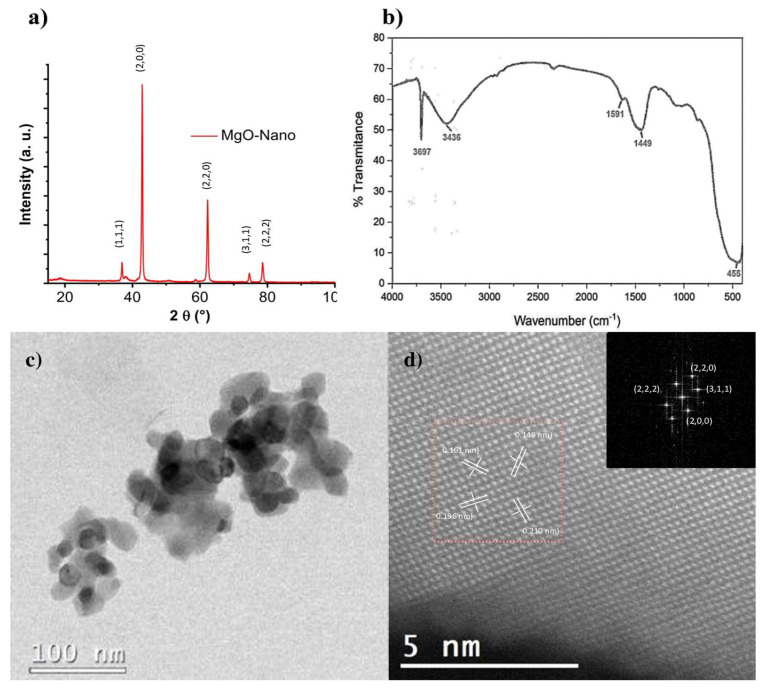
Characterization of MgO-NPs. (**a**) In X-ray diffraction, the peaks observed were indexed according to the JCPDS with the card number: 89-7746, and the corresponding crystal planes of MgO were indexed to (111), (200), (220), (311) and (222). (**b**) In FT-IR spectra, the signals at 3697 cm^−1^ and 3436 cm^−1^ are characteristic of the O–H stretching band. The peaks at 1449–1591 cm^−1^ correspond to the bending vibration of the O–H; these strong bands appear due to the hygroscopic nature of MgO, whereas the strong peak around 455 cm^−1^ is due to the stretching vibrations of the bond between Mg and O. (**c**) Transmission electron microscopy (high resolution) of MgO-NPs indicate the shapes to be polyhedric, with sizes smaller than or close to 100 nm. Likewise, the results of the analysis conducted in high resolution (**d**), showed that the interplanar distances coincide with the reflections of the crystal structure of the MgO.

**Figure 2 nanomaterials-13-00502-f002:**
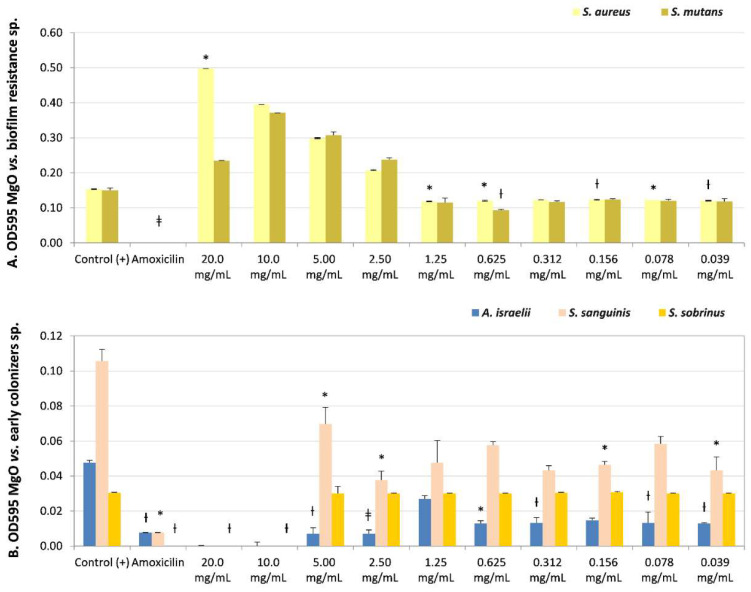
Microbial susceptibility of MgO-NPs against (**A**) resistant biofilm strains *Staphylococcus aureus* and *Streptococcus mutans*. (**B**) Early colonizers were *Actinomyces israelii, Streptococcus sanguinis,* and *Streptococcus sobrinus*. OD595: optical density λ at 595 nm averaged triplicated values; Control (+): bacteria positive control, Amoxicillin: inhibition control, and MgO dissolution used (20 mg/mL to 0.039 mg/mL). Paired differences were determined by Student’s *t*-test after adjusting per triplicate results as previously described by [27]. The difference between each strain Control + versus each plate dilution is shown as * *p <* 0.05; ƚ *p <* 0.01; ǂ *p <* 0.001.

**Figure 3 nanomaterials-13-00502-f003:**
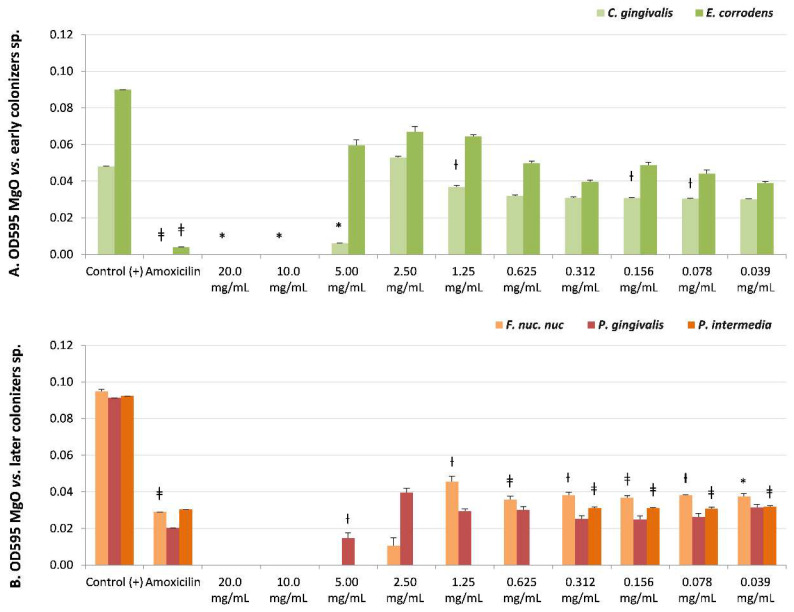
Microbial susceptibility of MgO-NPs against (**A**) early colonizer strains *Capnocytophaga gingivalis* and *Eikenella corrodens* and (**B**) later colonizers *F. nuc. nuc*: *Fusobacterium nucleatum* subsp. *nucleatum*, *Porphyromonas gingivalis*, and *Prevotella intermedia*. OD595: optical density λ at 595 nm averaged triplicated values; Control (+): bacteria positive control; Amoxicillin: inhibition control, and MgO dissolution used (20 mg/mL to 0.039 mg/mL). Paired differences were determined by Student’s *t*-test after adjusting per triplicate results as previously described by [27]. The difference between each strain Control + versus each plate dilution is shown as * *p* < 0.05; ƚ *p* < 0.01; ǂ *p* < 0.001.

**Table 1 nanomaterials-13-00502-t001:** Oral species evaluated.

Species	ATCC	Biofilm
*Actinomyces israelii*	12102	Early colonizers
*Capnocytophaga gingivalis*	33624	Early colonizers
*Eikenella corrodens*	23834	Early colonizers
*Staphylococcus aureus*	25923	Biofilm resistance *
*Streptococcus mutans*	25175	Biofilm resistance *
*Streptococcus sanguinis*	10556	Early colonizers
*Streptococcus sobrinus*	33478	Early colonizers
*Fusobacterium nucleatum* subsp. *nucleatum*	25586	Later colonizers
*Porphyromonas gingivalis*	33277	Later colonizers
*Prevotella intermedia*	25611	Later colonizers

ATCC: American Type Culture Collection (Rockville, MD); Biofilm: early and later oral biofilm colonizers of the subgingival dental plaque and * biofilm resistance strains of disposable medical devices.

**Table 2 nanomaterials-13-00502-t002:** Microbial susceptibility of biofilm former species under MgO-NPs.

Species	*A. israelii*			*C. gingivalis*			*E. corrodens*			*S. aureus*			*S. mutans*		
	OD595	*ST*	CFU	OD595	*TS*	CFU	OD595	*ST*	CFU	OD595	*ST*	CFU	OD595	*ST*	CFU
Control (+)	0.048	-	(+)	0.048	-	(+)	0.090	-	(+)	0.153	-	(+)	0.150	-	(+)
Amoxicillin	0.008	ƚ	0	0.000	ǂ	0	0.004	ǂ	0	0	ǂ	0	0	ǂ	0
20.0 mg/mL	0	-	0	0.000	*	0	0	-	0	0.498	*	15	0.235	-	0
10.0 mg/mL	0	-	0	0.000	*	0	0	-	0	0.396	-	2	0.371	-	4
5.00 mg/mL	0.007	ƚ	0	0.006	*	6	0.060	-	(+)	0.299	-	3	0.307	-	6
2.50 mg/mL	0.007	ǂ	0	0.053	-	(+)	0.067	-	(+)	0.208	-	2	0.237	-	6
1.25 mg/mL	0.027	-	0	0.037	ƚ	(+)	0.064	-	(+)	0.118	*	(+)	0.115	-	2
0.625 mg/mL	0.013	*	0	0.032	-	(+)	0.050	-	(+)	0.120	*	(+)	0.094	ƚ	0
0.312 mg/mL	0.013	ƚ	0	0.031	-	(+)	0.040	-	(+)	0.123	-	(+)	0.117	-	0
0.156 mg/mL	0.015	-	0	0.031	ƚ	(+)	0.049	-	(+)	0.123	ƚ	(+)	0.124	-	(+)
0.078 mg/mL	0.013	ƚ	0	0.030	ƚ	(+)	0.044	-	(+)	0.123	*	(+)	0.120	-	(+)
0.039 mg/mL	0.013	ƚ	0	0.030	-	(+)	0.039	-	(+)	0.120	ƚ	(+)	0.119	-	(+)
**Species**	* **S. sanguinis** *			* **S. sobrinus** *			* **F. nuc. nuc** *			* **P. gingivalis** *			* **P. intermedia** *		
	**OD595**	* **ST** *	**CFU**	**OD595**	* **ST** *	**CFU**	**OD595**	* **ST** *	**CFU**	**OD595**	* **ST** *	**CFU**	**OD595**	* **ST** *	**CFU**
Control (+)	0.106	-	(+)	0.030	-	(+)	0.095	-	(+)	0.091	-	(+)	0.092	-	(+)
Amoxicillin	0.008	*	0	0	ƚ	0	0.029	ǂ	0	0.020	-	0	0.030	-	0
20.0 mg/mL	0.000	-	0	0	ƚ	0	0	-	0	0	-	0	0	-	0
10.0 mg/mL	0.000	-	(+)	0	ƚ	0	0	-	0	0	-	1	0	-	0
5.00 mg/mL	0.070	*	(+)	0.030	-	0	0	-	0	0.015	ƚ	10	0	-	0
2.50 mg/mL	0.038	*	(+)	0.030	-	3	0.011	-	10	0.040	-	300	0	-	0
1.25 mg/mL	0.048	-	(+)	0.030	-	(+)	0.046	ƚ	(+)	0.029	-	(+)	0	-	0
0.625 mg/mL	0.058	-	(+)	0.030	-	(+)	0.036	ǂ	(+)	0.030	-	(+)	0	-	0
0.312 mg/mL	0.043	-	(+)	0.030	-	(+)	0.038	ƚ	(+)	0.025	-	(+)	0.031	ǂ	(+)
0.156 mg/mL	0.046	*	(+)	0.031	-	(+)	0.037	ǂ	(+)	0.025	-	(+)	0.031	ǂ	(+)
0.078 mg/mL	0.058	-	(+)	0.030	-	(+)	0.038	ƚ	(+)	0.026	-	(+)	0.031	ǂ	(+)
0.039 mg/mL	0.043	*	(+)	0.030	-	(+)	0.037	*	(+)	0.031	-	(+)	0.032	ǂ	(+)

OD595: optical density λ at 595 nm averaged triplicate values; *ST:* Student’s *t*-test *; CFUs: colony-forming units (1:1 dissolution mg/mL); minimum inhibitory concentration was determined by the CFUs (pink boxes). Minimum bactericidal concentration was determined by a lack of growing colonies (red boxes), and resistance strains against MgO-NPs were determined as “+” or uncountable and growing. *F. nuc. nuc: Fusobacterium nucleatum* subsp. *nucleatum.* * Paired differences were determined after adjusting statistics as previously described in [27]. Paired differences were between each positive bacteria “Control (+)” vs. each averaged triplicated dissolution: * *p* < 0.05; ƚ *p* < 0.01; ǂ *p* < 0.001.

## Data Availability

The authors declare that they have followed the protocols of the School of Dentistry of the National Autonomous University of Mexico (UNAM). The authors are the owners of this document, as well as the data results when requested

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
