# Peer review of "Antibacterial Properties In Vitro of Magnesium Oxide Nanoparticles for Dental Applications"

_nanomaterials, 2023, doi:10.3390/nano13030502_

Round 1
Reviewer 1 Report
The authors have prepared MgO-Nano as an antibacterial agent against representative oral bacteria. MgO-Nano with cubic crystal structure has been obtained by magnesium hydroxide activation and then has been annealed at 800°C. The structural properties of MgO-Nano have been demonstrated by XRD and HR-TEM. The authors have demonstrated that MgO-Nano has revealed promising antibacterial activity in all oral biofilm strains. Overall, this work can inspire more material design ideas of MgO-based nanomaterials for antibacterial application in dental. Therefore, I would like to recommend this work to publish in Nanomaterials. Below are some comments for the authors.
1. The common abbreviation of nanoparticles is NPs. I suggest the authors to replace the abbreviation of Nono with NPs.
2. The description of Figure 1a-d should be added in the manuscript.
3. In Figure 1a, the characteristic peaks of MgO-Nano should be labeled in the XRD spectrum. The description should be also added in the main text.
4. In FTIR spectrum of Figure 1b, the authors indicated that the signals at 3697 cm-1 and 3436 cm-1 are characteristic of the O-H stretching band. What is the reason? Generally, there is only one broad O-H stretching band.
5. For TEM image of Figure 1c, please remove the measuring lines and numbers.
6. For HR-TEM image of Figure 1d, please measure the lattice distance to demonstrate the crystal plane of MgO-Nano.
7. For the introduction “Other findings indicate the potential number of reactive groups dependent on superficial oxygen and defect points on their surface”, more references could be cited to broaden the introduction.
https://doi.org/10.2147/IJN.S328767
Author Response
Response to Reviewer 1 Comments
As noted, and in agreement with Reviewer 1 we do the corresponding changes as follows:
- The common abbreviation of nanoparticles is NPs. I suggest the authors to replace the abbreviation of Nono with NPs.
Response 1: We had already written this point with the correspondent changes in all the manuscript sections with NPs instead of Nano. You can check this with the word tracker tool.
- The description of Figure 1a-d should be added in the manuscript.
Response 2: We added the specifications on page: 5, lines:186-202.
- In Figure 1a, the characteristic peaks of MgO-Nano should be labeled in the XRD spectrum. The description should be also added in the main text.
Response 3: To clarify this point in the manuscript, we add the following paragraph on the page 4, lines 186 – 190:
The X-ray diffraction patterns obtained from the MgO-NPs show a face-centered cubic (fcc) lattice with the space group Fm3m (Figure 1a). The peaks observed were indexed according to the Joint Committee on Powder Diffraction Standards (JCPDS) with the card number: 89-7746, and the corresponding crystal planes of MgO were indexed (111), (200), (220), (311) and (222).
- In FTIR spectrum of Figure 1b, the authors indicated that the signals at 3697 cm-1 and 3436 cm-1 are characteristic of the O-H stretching band. What is the reason? Generally, there is only one broad O-H stretching band.
Response 4: To respond to this comment, in the case of these nanoparticles with this methodology, the appearance of these signals in the table of IR absorptions for representative functional groups was corroborated.
- For TEM image of Figure 1c, please remove the measuring lines and numbers.
Response 5: In agreement to this comment, we replacement the figure 1c and remove the measuring lines and numbers.
- For HR-TEM image of Figure 1d, please measure the lattice distance to demonstrate the crystal plane of MgO-Nano.
Response 6: In agreement to this observation, we replacement the figure 1d and added the measure the lattice distance.
- For the introduction “Other findings indicate the potential number of reactive groups dependent on superficial oxygen and defect points on their surface”, more references could be cited to broaden the introduction. https://doi.org/10.2147/IJN.S328767
Response 7: As noted and in agreement with Reviewer 1, we clarify add the recommended reference [2] in page 2, line 58.
Reviewer 2 Report
The manuscript is interesting and informative with interesting results. However, the written English needs to be revised as there are quite a few grammatical errors or sentences that do not make much sense (if read as is). Furthermore, some terms are incorrect, e.g. what is a T-Student test? (It is Student's t-test).
Author Response
Response to Reviewer 2 Comments
The manuscript is interesting and informative with interesting results. However, the written English needs to be revised as there are quite a few grammatical errors or sentences that do not make much sense (if read as is). Furthermore, some terms are incorrect, e.g. what is a T-Student test? (It is Student's t-test).
Response: The manuscript was revised and corrected by a professional interpreter and the English corrections are tracked with the corresponding word tool. Particularly we correct Student's t-test instead of T-Student test on the abstract (line 30) and on pages: 4 (line 172), 5 (line 175), page 6 (line 244), page 7 (line 249), page 8 (line 267), page 9 (280), and on table 1 labels.
Reviewer 3 Report
My comments are in attached file.

Author Response
Response to Reviewer 3 Comments
After reading of the first page of the manuscript, it is clear that the text is not ready for scientific analysis. Practically, each sentence is unclear or contains evident defects. In my opinion, initially, English should be tested and corrected by professional interpreter. Also, it is topical to use conventional scientific terminology. Then, the paper could be resubmitted and considered for review. My several corrections spontaneously proposed for the text are listed below for author consideration.
The authors are grateful for the Reviewer 3 observations and we appreciate the corrections. The manuscript was improved by a professional interpreter and the English corrections are tracked with the corresponding word tool. Additionally, in order to improve on the clarity and readability of our manuscript we have added some corrections as the follow comments:
Page 1
- Oral bacteria have been considered biofilm-resistance strains related to commensal infections such as dental caries, periodontitis, or peri-implantitis.
This sentence is unclear.
Page 1
- However, their antibacterial properties have been poorly studied.
However, their antibacterial properties are poorly studied.
Page 1
- Considering this, the purpose of this investigation was to determine the antibacterial properties of MgO-Nano, on representative oral strains.
Considering this, the purpose of this investigation was to determine the antibacterial properties of MgO-Nano on representative oral strains.
Response 1-3: Corrections in page 1 lines 12-20:
Dental caries, periodontitis, or peri-implantitis are commensal infections related to oral biofilm former bacteria. Applications of nanoparticles of Magnesium oxide nanoparticles (MgO-NPs.) were studied related to the antibacterial properties of a few microorganisms. Considering this the purpose of the present investigation was to determine the antibacterial properties of MgO-NPs on representative oral strains.
Page 1
- MgO-Nano with cubic crystal structure had obtained by magnesium hydroxide activation. After grinding MgO-Nano product was annealed at 800°C (2 h). MgO-Nano obtained were tested against ten oral ATCC strains at ten serial dissolutions (1:1_20.0-0.039mg/mL per triplicated) using Micro-Broth dilution method and reporting Minimal-Inhibitory-Concentration_(MIC) or Minimal-Bactericidal-Concentration_(MIB).
MgO-Nano with cubic crystal structure was obtained by magnesium hydroxide activation. After grinding, MgO-Nano product was annealed at 800°C (2 h). MgO-Nano obtained was tested against ten oral ATCC strains at ten serial concentrations (1:1_20.0-0.039mg/mL per triplicated) using Micro-Broth dilution method to determine Minimal-Inhibitory-Concentration_(MIC) or Minimal-Bactericidal-Concentration_(MIB).
Response 4: Corrections in page 1 lines 22-29:
MgO-NPs obtained were tested against ten oral ATCC strains at ten serial concentrations_(1:1_20.0-0.039mg/mL per triplicated) using Micro-Broth dilution method to determine Minimal-Inhibitory-Concentration_(MIC) or Minimal-Bactericidal-Concentration_(MIB).
Page 1
- The crystal form structures, and average sizes were 21 nm, by X ray diffraction and Transmission-Electron-Microscopy_(High-Resolution).
The crystal form structures, and average size was 21 nm, as determined by X ray diffraction and Transmission-Electron-Microscopy_(High-Resolution).
This fragment is unclear: "The crystal form structures".
Response 5: Corrections in page 1 lines 31-33:
The structure of the crystal form and the average size was 21 nm, as determined by X-ray diffraction and Transmission-Electron-Microscopy_(High-Resolution).
Page 1
- The sensitivity observed of Streptococcus sanguinis_(MIB_20mg/mL), Capnocytophaga gingi-24 valis_(MIB/MIC_10-5mg/mL), and Eikenella corrodens_(MIB_10mg/mL) at high concentrations and, at lower concentrations of Actinomyces israelii_(MIB_0.039mg/mL), Staphylococcus aureus_(MIC_2.5mg/mL), Streptococcus mutans_(MIB_20 mg/mL/MIC_0.321mg/mL), Streptococcus sobrinus_(MIB/MIC_5-2.5mg/mL), Fusobacterium nucleatum subsp. nucleatum_(MIB/MIC_5-2.5mg/mL), Prevotella intermedia_(MIB_0.625mg/mL), and Porphyromonas gingivalis_(MIB_20 29 mg/mL/MIC_2.5mg/mL),
This sentence is unclear.
Response 6: Corrections in page 1 lines 34-48:
Antimicrobial sensitivity was observed in Streptococcus sanguinis_(MIB_20mg/mL), Capnocytophaga gingivalis_(MIB/MIC_10-5mg/mL), and Eikenella corrodens_(MIB_10mg/mL) at high concentrations of the MgO-NPs At lower concentrations of the MgO-NPs, sensitivity was observed in Actinomyces israelii_(MIB_0.039mg/mL), Staphylococcus aureus_(MIC_2.5mg/mL), Streptococcus mutans_(MIB_20mg/mL/MIC_0.321mg/mL), Streptococcus sobrinus_(MIB/MIC_5-2.5mg/mL), Fusobacterium nucleatum subsp. nucleatum_(MIB/MIC_5-2.5mg/mL), Prevotella intermedia_(MIB_0.625mg/mL), and Porphyromonas gingivalis_(MIB_20 mg/mL/MIC_2.5mg/mL).
Page 1
- MgO-Nano evaluated had reported selective antibacterial in all oral biofilm strains evaluated with potential use in dental applications.
This sentence is unclear.
Response 7: Corrections in page 1 lines 48-51:
MgO-NPs evaluated were reported antibacterial sensitivity in all oral biofilm strains evaluated with potential use in dental applications.
Page 1
- The use of magnesium oxide as a nanomaterial (MgO-Nano) has increased, especially in the medical field such as biomolecular detection, diagnostics, and microelectronics.
The use of magnesium oxide nanomaterial s(MgO-Nano) is increased, especially in the medical field such as biomolecular detection, diagnostics, and microelectronics.
What is the reason to classify microelectronics as a part of medical field?
Response 8: The reason is this technology keeps highest levels for high-throughput screening (HTS) methods for DNA sequencing, in which nano biochips are mainly used to stretch DNA strands. Similarly, nanofluidic devices are strategically targeted to be used as disposable devices, specially applied to point-of-care (POC) diagnostics. To be clearer we added the following statement page 1, lines 38-39.
The use of magnesium oxide as a nanomaterial (MgO-NPs) has increased, especially in the medical field such as diagnostics, detection and biosensors to molecular behavior [1, 2].
The corresponding references to support this:
[2] Bhattarai, P., & Hameed, S. (2020). Basics of biosensors and nanobiosensors. Nanobiosensors: From Design to Applications, 1-22.
Page 1
- The antibacterial effect of nanoparticles based on magnesium oxide has shown antimicrobial properties on a wide range of microorganisms, where the antibacterial effectivity is due to the size [1, 2] and dosage dependence [3, 4].
This sentence is unclear. How is it possible: "The antibacterial effect" ... "has shown antimicrobial properties"?!
Response 9: To clarify this paragraph we correct it as follows on page 2 (lines 58-64):
The antibacterial effect of nanoparticles based on magnesium oxide has shown effective properties on a wide range of microorganisms, where the antibacterial mechanism of action is due to the size [2, 3] and dosage dependence [4, 5].
Page 1
- Regarding the antibacterial mechanism of MgO-Nano there are different approximations to explain it, such as the interaction of nanoparticles with bacteria, such as the production of reactive oxygen species, concentrations of Ca2+ ion, and specific molecular sites contribute to the mechanisms of
action by MgO-Nano against bacteria in the platonic state but increase the pH of 7 to 10 and the concentrations of Mg2+ ion.
This sentence is unclear. What is the nature of Ca2+ ions in MgO? What is a "platonic state" of bacteria? Howis it possible to change concentration of Mg2+ ions in MgO?
Response 10: To clarify the reviewer questions:
- The nature of Ca2+ is inside bacteria cells.
- A planktonic state of bacteria is the same that free-living bacteria.
- Concentrations of Mg2+ ions can change inside bacteria cells in the presence of MgO-NPs.
The corresponding references to support this: https://www.mdpi.com/2073-4344/11/7/821,
- Al-Hazmi, F.; Alnowaiser, F.; Al-Ghamdi, A.A.; Al-Ghamdi, A.A.; Aly, M.M.; Al-Tuwirqi, R.M.; El-Tantawy, F. A new large-scale synthesis of magnesium oxide nanowires: Structural and antibacterial properties. Superlattices Microstruct. 2012, 52, 200–209. [Google Scholar] [CrossRef][Green Version]
- Karthik, K.; Dhanuskodi, S.; Gobinath, C.; Prabukumar, S.; Sivaramakrishnan, S. Fabrication of MgO nanostructures and its efficient photocatalytic, antibacterial and anticancer performance. J. Photochem. Photobiol. B Biol. 2019, 190, 8–20. [Google Scholar] [CrossRef] [PubMed]
- Wang, L.; Hu, C.; Shao, L. The antimicrobial activity of nanoparticles: Present situation and prospects for the future. Int. J. Nanomed. 2017, 12, 1227–1249. [Google Scholar] [CrossRef][Green Version]
Corrections in page 2 lines 64-70:
Regarding the antibacterial mechanism of MgO-NPs, there are different approximations to explain it. The interaction of nanoparticles with bacteria increases the production of reactive oxygen species such as the concentrations of Ca2+ ions inside bacteria cells. Additionally, the union of MgO-NPs, to specific molecular sites of bacteria in the planktonic state can trigger membrane disruptions leading to bacteria death and the cytoplasm bacteria alkalinization of pH from 7 to 10 with high concentrations of Mg2+ ions [5, 6].
Also, to improve refences, we decided to add the follow bibliography:
[6] Wang, L.; Hu, C.; Shao, L. The antimicrobial activity of nanoparticles: Present situation and prospects for the future. Int. J. Nanomed. 2017, 12, 1227–1249. [Google Scholar] [CrossRef][Green Version]
Page 1
- Other findings indicate the potential number of reactive groups dependent on superficial oxygen and defect points on their surface [2, 4].
This is a mixed sentence without clear subject.
Response 11: To clarify the paragraph corrections are in page 2 lines 75-77:
Other mechanism actions of MgO-NPs indicate the potential number of reactive groups dependent on superficial oxygen and defect points on the bacteria surfaces.
Page 2
- Despite these studies, and unlike other metal oxide nanoparticles, there is not so much information of MgO nanoparticles for biological and antimicrobial applications [1, 5].
This is first clear sentence.
Response 12: We want to thank you for the reviewer 3 comments to correct the manuscript.
Round 2
Reviewer 1 Report
- The description of Figure 1a-d should be added in the manuscript.
Response 2: We added the specifications on page: 5, lines:186-202.
I cannot find the description on 186-202.
- In FTIR spectrum of Figure 1b, the authors indicated that the signals at 3697 cm-1 and 3436 cm-1 are characteristic of the O-H stretching band. What is the reason? Generally, there is only one broad O-H stretching band.
Response 4: To respond to this comment, in the case of these nanoparticles with this methodology, the appearance of these signals in the table of IR absorptions for representative functional groups was corroborated.
At least, please cite the related references.
- For the introduction “Other findings indicate the potential number of reactive groups dependent on superficial oxygen and defect points on their surface”, more references could be cited to broaden the introduction. https://doi.org/10.2147/IJN.S328767
Response 7: As noted and in agreement with Reviewer 1, we clarify add the recommended reference [2] in page 2, line 58.
The authors have cited in wrong sentence and reference.
Reviewer 3 Report
My coments are attached.

Round 3
Reviewer 1 Report
The revision has been improved. Therefore, this manuscript can be published in Nanomaterials as its current form.
Reviewer 3 Report
After the revision, this paper is sufficiently improved, and it can be published in the present state.